# Cofactor Engineering for Efficient Production of α-Farnesene by Rational Modification of NADPH and ATP Regeneration Pathway in *Pichia pastoris*

**DOI:** 10.3390/ijms24021767

**Published:** 2023-01-16

**Authors:** Sheng-Ling Chen, Ting-Shan Liu, Wei-Guo Zhang, Jian-Zhong Xu

**Affiliations:** The Key Laboratory of Industrial Biotechnology, Ministry of Education, School of Biotechnology, Jiangnan University, 1800# Lihu Road, Wuxi 214122, China

**Keywords:** α-farnesene, *Pichia pastoris*, cofactor engineering, pentose phosphate pathway, NADH kinase

## Abstract

α-Farnesene, an acyclic volatile sesquiterpene, plays important roles in aircraft fuel, food flavoring, agriculture, pharmaceutical and chemical industries. Here, by re-creating the NADPH and ATP biosynthetic pathways in *Pichia pastoris*, we increased the production of α-farnesene. First, the native oxiPPP was recreated by overexpressing its essential enzymes or by inactivating glucose-6-phosphate isomerase (PGI). This revealed that the combined over-expression of ZWF1 and SOL3 increases α-farnesene production by improving NADPH supply, whereas inactivating PGI did not do so because it caused a reduction in cell growth. The next step was to introduce heterologous cPOS5 at various expression levels into *P. pastoris.* It was discovered that a low intensity expression of cPOS5 aided in the production of α-farnesene. Finally, ATP was increased by the overexpression of APRT and inactivation of GPD1. The resultant strain *P. pastoris* X33-38 produced 3.09 ± 0.37 g/L of α-farnesene in shake flask fermentation, which was 41.7% higher than that of the parent strain. These findings open a new avenue for the development of an industrial-strength α-farnesene producer by rationally modifying the NADPH and ATP regeneration pathways in *P. pastoris.*

## 1. Introduction

α-Farnesene, one of the simplest sesquiterpenes, has an enormous application in nature and industry. For example, α-farnesene works as a chemical signaling molecule to signal danger and to implicate the orientation of aphids and termites in nature [1]. Additionally, α-farnesene serves as an intermediate in the industry’s production of high-value products such as squalane, biofuel, vitamin E and vitamin K1 [2,3,4]. The use of α-farnesene in agriculture, chemicals, bioenergy, medicine and cosmetics is therefore significant economically [1,3]. Plant extraction is the main way to make α-farnesene since it is widely present in plants, such as apples and *Artemisia annua* [1,4,5]. However, the drawbacks of plant extraction, such as poor yield, high production costs, the scarcity of feedstock and the severe environmental damage, restrict their industry use [3,5,6]. Researchers focus on using microbial fermentation to produce α-farnesene as a result, and numerous efficient methods have been used to modify microorganisms to enhance the biosynthesis of α-farnesene, such as enhancing the α-farnesene biosynthesis pathway, blocking the downstream α-farnesene biosynthesis pathway, rewriting the central carbon metabolism, compartmentalizing the supply ways of precursors, relieving the inhibition of cell growth and optimizing the metabolic process [3,4,7,8,9]. In the previous study, we constructed an α-farnesene high-producing strain, *Pichia pastoris* X33-30, by the dual regulation of cytoplasm and peroxisomes [3]. P_CAT1_ promoters were replaced with P_GAP_ promoters in X33-30 to obtain the strain X33-30*, which produced 2.17 ± 0.15 g/L of α-farnesene in shake flasks. The strain *P. pastoris* X33-30* enhanced the supply of isopentenyl pyrophosphate (IPP) and dimethylallyl pyrophosphate (DMAPP). Despite the fact that there are two pathways for producing IPP and DMAPP, the mevalonate (MVA) pathway and the methylerythritol-4-phosphate (MEP) pathway [5], no fewer than six molecules of ATP and NADPH are required to produce one molecule of α-farnesene (Figure 1). The overall stoichiometry of α-farnesene biosynthesis via the MVA pathway is: 9 acetyl-CoA + 9 ATP + 3 H_2_O + 6 NADPH + 6 H^+^ → 1 α-farnesene + 9 CoA + 6 NADP^+^ + 9 ADP + 3 Pi + 3 PPi + 3 CO_2_ [10]. With the exception of the precursor acetyl-CoA, the α-farnesene synthesis equation shows that the cofactors ATP and NADPH are also crucial. However, relatively little research has focused on the role of ATP and NADPH in the synthesis of α-farnesene.

NADPH acts as a cofactor for catalyzing the formation of mevalonate from 3-hydroxy-3-methyl glutaryl coenzyme A (i.e., HMG-CoA) (Figure 1). Moreover, the extra demand for NADPH has been discussed as being responsible for the heterologous protein production. Researchers have found that the amount of NADPH available is closely linked to the amount of biomass and heterologous proteins [11,12]. In addition, NADPH is also used to protect cells against endoplasmic reticulum (ER) stress and oxidative stress [11,13]. It should be noted that the MVA pathway is the major pathway for producing IPP and DMAPP in *P. pastoris* [3], and thus six molecules of NADPH and nine molecules of ATP are needed to produce one molecule of α-farnesene. On the other hand, yeast and bacteria use NADH more than NADPH as a reduced cofactor of catabolism [4]. Thus, increasing the intracellular NADPH level or eliminating NADPH consumption is a common strategy to facilitate NADPH-dependent products, including terpenoid [14,15]. For example, Liu et al. compared the effects of six native enzymes involved in NAPDH regeneration in *Yarrowia lipolytica* and found that mannitol dehydrogenase benefits the increase in squalene production [16]. In addition, the introduction of a synthetic version of the Entner–Doudoroff pathway from *Zymomonas mobilis* into *E. coli* MG1655 has been shown to be able to increase the NADPH regeneration rate 25-fold and thus increase terpenoid production [17]. In *P. pastoris*, there are two inherent routes for NADPH generation, i.e., the oxidative branch of the pentose phosphate pathway (oxiPPP) and the acetate biosynthetic pathway [9,11,18]. However, the key enzymes in oxiPPP were negatively controlled by NADPH and ATP at the transcriptional and/or the translational level [11,19]. Although the heterogeneous expression of NADH kinase (i.e., POS5, catalyzed NADH to form NADPH) would increase the NADPH regeneration in *P. pastoris* [11], NADH plays a pivotal role in ATP regeneration [18,20]. Except as the energy currency in cells, ATP is a key factor in α-farnesene biosynthesis (Figure 1) [21]. Therefore, ATP availability is extremely important for cell growth and α-farnesene biosynthesis and adequate supplies of NADH are needed for the cell because ATP is mainly produced by NADH oxidation via electron transport phosphorylation (ETP) under aerobic conditions [22]. How to efficiently supply NADPH and ATP has already become an important research direction in developing an α-farnesene high-producing strain. 

In this work, the biosynthetic pathways of NADPH and ATP were rationally reconstructed in a strain of *P. pastoris* that produces a lot of α-farnesene, called *P. pastoris* X33-30*. This reconstructed the metabolic pathways of carbon in *P. pastoris* X33 to produce more α-farnesene. To do this, the native oxiPPP was first rebuilt in the strain X33-30* by over-expressing the key enzymes in oxiPPP or by turning off the glucose-6-phosphate isomerase in glycolysis. Subsequently, the heterologous POS5 from *S. cerevisiae* was introduced into *P. pastoris* and controlled by different intensities of promoters to further optimize the NADPH supply. Finally, increasing ATP availability was attempted by increasing the supply of adenosine monophosphate (AMP) for ATP synthesis and decreasing NADH consumption in the shunt pathway. As a result, the resultant strain *P. pastoris* X33-38 produced 3.09 ± 0.37 g/L of α-farnesene after 72 h in shake flask fermentation. These findings show that increasing the availability of NADPH and ATP in *P. pastoris* increases α-farnesene production and offers a new way to build an industrial-strength α-farnesene producer by rationally modifying the NADPH and ATP regeneration pathway in *P. pastoris*.

## 2. Results and Discussion

### 2.1. Combined Over-Expression of ZWF1 and SOL3 Improves the NADPH Supply and Thus Increases α-Farnesene Production in P. pastoris X33-30*

The oxiPPP is the main inherent route for NADPH generation in *P. pastoris*, which is catalyzed by glucose-6-phosphate dehydrogenase (ZWF1), 6-gluconolactonase (SOL3), 6-phosphogluconate dehydrogenase (GND2), and D-ribulose-5-phosphate 3-epimerase (RPE1) [11,18]. The key enzymes in oxiPPP were optimized to overexpress in a α-farnesene high-producing strain, *P. pastoris* X33-30*, to increase NADPH availability for α-farnesene. Firstly, we overexpressed the single genes *zwf1* (encoding ZWF1), *sol3* (encoding SOL3), *gnd2* (encoding GND2) and *rpe1* (encoding RPE1) in strain X33-30*, which resulted in strains X33-30*Z, X33-30*S, X33-30*G and X33-30*R, respectively. Compared to strain X33-30*, strains X33-30*Z and X33-30*S had higher NADPH concentrations, whereas strains X33-30*G and X33-30*R had no discernible difference in NADPH concentrations at 24 and 72 h (Figure 2a). Similarly, strains X33-30*Z and X33-30*S increased α-farnesene production by about 8.7% and 12.9%, respectively, when compared to strain X33-30* at 72 h (Figure 2b).The similar results were also found in previous studies, in which the overexpression of ZWF1 or SOL3 increased the foreign protein production in *P. pastoris* [23,24,25] and the terpenoid production in *S. cerevisiae* [26,27] due to the increased intracellular NADPH level. ZWF1 and SOL3 catalyzed the first and the second steps of the oxiPPP and were inhibited by NADPH and ATP [11,23], which catalyzed the rate-limiting steps in oxiPPP [24]. In addition, the native expression level of *sol3* in *P. pastoris* was low [28]. Therefore, this may be why the strain X33-30*Z with the overexpression of ZWF1 and the strain X33-30*S with the overexpression of SOL3 increased the NADPH availability and thus increased α-farnesene production. It should be noted that the overexpression of GND2 had no positive effects on increasing the NADPH availability and α-farnesene production (Figure 2), which was similar to the results reported by Kim et al. [29] and Nocon et al. [24] but different to the results reported by Prabhu and Veeranki [23]. Rebnegger et al. [28] pointed out that *gnd2* shows a high expression level, while *sol3* shows a low expression level in *P. pastoris*. Based on this, we speculated that GND2 was not a rate-limiting enzyme and the overexpression of GND2 only did not enhance the carbon flux in oxiPPP.

To further analyze the synergetic effects of these key genes in oxiPPP on NADPH and α-farnesene production, we tried out different expression combinations of these genes, looking for the highest performing combos. As can be seen from Figure 3a, the resulting strain X33-30*ZSGR (i.e., the combined overexpression of ZWF1, SOL3, GND2 and RPE1) showed the highest intracellular NADPH level, followed by the strain X33-30*ZSG (i.e., the combined overexpression of ZWF1, SOL3 and GND2). Interestingly, the combined overexpression of GND2 and RPE1 (i.e., strain X33-30*GR) had no obvious effect on increasing NADPH levels (Figure 3a), which was similar to the single overexpression of GND2 or RPE1 (Figure 2a). This may be due to the bottlenecks of the rate-limiting step, which is catalyzed by ZWF1 and SOL3 [24]. However, it should be noted that α-farnesene production was not increased with the increase in the intracellular NADPH level in the corresponding strain (Figure 3b). Among the test strains, the strain X33-30*ZS (i.e., strain X33-31) produced the highest α-farnesene (i.e., 2.54 ± 0.21 g/L) despite the third-highest NADPH levels (Figure 3). In contrast, although the strains X33-30*ZSGR and X33-30*ZSG showed the top two NADPH levels (Figure 3a), they showed the worst α-farnesene production, even lower than the original strain X33-30* (Figure 2b and Figure 3b). Nocon et al. [24] and Prabhu and Veeranki [23] also found similar results, in which the combined overexpression of ZWF1, SOL3, RPE1 or/and GND2 would be detrimental to foreign protein production. This may be due to the fact that the excess overexpression of the key enzymes in oxiPPP may imbalance the PPP flux [24] or disturb the acetyl-CoA biosynthetic pathway [29], thus negatively impacting product formation. As can be seen from Figure 2 and Figure 3, the NADPH content is lower at 72 h than at 24 h, most probably because the α-farnesene synthesis requires a large amount of NADPH in the middle and late stages of fermentation and thus consumes a large amount of intracellular NADPH. Although the α-farnesene yield of strain X33-30*S and strain X33-31 did not show a statistically significant difference in Figure 3b (data not shown), the α-farnesene yield of strain X33-31 was higher than that of strain X33-30*S (Figure 3b). In order to finally obtain the highest α-farnesene producing strain, the strain X33-31 was chosen to be further modified to increase α-farnesene production.

### 2.2. Inactivation of Glucose-6-Phosphate Isomerase Disturbs the Cell Growth and Thus Negatively Affects α-Farnesene Biosynthesis in P. pastoris X33-31

Previous research indicated that the overexpression of the transcription factor STB5 increased cytosolic NADPH concentrations because STB5 upregulated the expression of most genes in the PPP and repressed the expression of glucose-6-phosphate isomerase-coding genes in glycolysis [29,30]. Due to the fact that STB5 is a basal regulator of the PPP, it is possible that the overexpression of STB5 did not increase protopanaxadiol production [29]. PGI (encoded by *pgi*) competes with ZWF1 for glucose-6-phosphate, which catalyzes glucose-6-phosphate to form frucose-6-phosphate (Figure 1). In order to investigate the effect of PGI on α-farnesene production, the PGI was inactivated in strain X33-31, resulting in strain X33-31ΔP. As a control, strain X33-30*ΔP (i.e., the deletion of *pgi* in strain X33-30*) was also achieved. Unfortunately, the inactivation of PGI in the strain X33-31 had negative effects on α-farnesene production, in which the resultant strain X33-31ΔP only produced about 5% of α-farnesene compared with the strain X33-31 (0.13 ± 0.06 g/L vs. 2.54 ± 0.21 g/L) (Figure 4a). In addition, the inactivation of PGI increased the intracellular NADPH level but repressed cell growth (Figure 4b,c). In the past, Qin et al. [31] also found that the expression of *pgi* controlled by the ultra-low intensity promoter NAT2p in *Saccharomyces cerevisiae* decreased the cell growth and 3-hydroxypropionic acid production. The possible reason for this could be that PGI plays an important role in the central carbon metabolism in yeast [32]. In addition, Aguilera and Zimmermann [33] suggested that the inactivation of PGI in *S. cerevisiae* prevents growth on glucose. However, it should be noted that turning off PGI in the strain X33-30* had little effect on producing more α-farnesene, even though cell growth slowed down (Figure 4). These results indicate that the inactivation of PGI enforced the carbon flux into PPP, thus increasing the NADPH availability for α-farnesene biosynthesis. Since the surplus NADPH cannot be re-oxidized, the PGI-deficient strain did not grow on glucose [34]. Thus, we speculated that the reason for which the strain X33-30*ΔP did not visibly decrease cell growth is that more NADPH was used to biosynthesize α-farnesene. The previous results have reinforced this speculation. For example, Fiaux et al. [35] restored cell growth on glucose of the PGI-deficient *S. cerevisiae* mutant through the heterologous expression of transhydrogenase UdhA from *E. coli*. Although the strain X33-30*ΔP produced more α-farnesene than the strain X33-30*, its final titer of α-farnesene was still lower than that of strain X33-31 (2.23 ± 0.18 g/L vs. 2.54 ± 0.21 g/L) (Figure 4a), indicating that inactivating the glucose-6-phosphate isomerase is not the best strategy for increasing α-farnesene production in strain X33-31.

### 2.3. Low Intensity Expression of POS5 from S. cerevisiae Balances the NADPH/NADH Ratio and Thus Promotes α-Farnesene Biosynthesis in P. pastoris

It is well known that the intracellular NADH level is higher than the intracellular NADPH level [36,37,38]. Previous research indicated that the heterologous expression of NADH kinase POS5 provides another source of NADPH, except for the oxiPPP in yeast [11,26]. To further promote α-farnesene production by optimizing the NADPH supply, we introduced the cPOS5 targeting in the cytosol from *S. cerevisiae* in strain X33-31. Interestingly, the resultant strain X33-32 with gene *cPOS5* controlled by promoter P_GAP_ showed bad cell growth and α-farnesene production, but it showed the increased productivity of NADPH and α-farnesene (Figure 5). POS5 catalyzed the NADH to form NADPH, thereby reducing cell energy resources [11]. In addition, excess NADPH in cell would repress cell growth, glucose consumption and product production [37,39]. These observations may be an underlying cause of the decreased cell growth and α-farnesene production. 

To try to solve this problem, we then decreased the expression level of POS5 by replacing P_GAP_ with a series of weak promoters. According to previous reports, the relative intensities of the promoters P_PISI_, P_GPM1_, P_MET3_ and P_PGK1_ were 40%, 15~40%, 13% and 0~10%, respectively, when compared to P_GAP_ [40,41,42]. α-Farnesene production (i.e., 2.77 ± 0.18 g/L) increased by about 9.1% in strain X33-35 with gene *cPOS5* controlled by P_MET3_ compared to strain X33-31 (i.e., 2.54 ± 0.21 g/L) (Figure 5c). Correspondingly, strain X33-35 also exhibited a high intracellular NADPH level (Figure 5b). In addition, decreasing the expression level of *POS5* restored cell growth as compared with the strain X33-32 (Figure 5a), indicating that excess NADPH in cells would be detrimental to cell growth. There results indicate that the overexpression of POS5 under the P_MET3_ promoter has a positive effect on promoting α-farnesene production. To further prove this conclusion, we overexpressed POS5 under the P_MET3_ promoter in the X33-30* strain, which resulted in strain X33-30*C. Strain X33-30*C showed a higher α-farnesene production and productivity than strain X33-30* because it exhibited a high intracellular NADPH level (Figure 5).

These results indicate that a low intensity expression of cPOS5 in strain X33-31 or strain X33-30* benefits from maintaining the right amount of NADPH for cell growth and α-farnesene production. Despite having a higher NADPH concentration and cell growth (Figure 5a,b), strain X33-34 produced no more α-farnesene than strain X33-31 (i.e., 2.56 ± 0.26 g/L vs. 2.54 ± 0.21 g/L) (Figure 5c), indicating that another limiting factor hampered α-farnesene biosynthesis in strain X33-34. As can be seen from Figure 1, nine molecules of ATP are needed to produce one molecule of α-farnesene. ATP is mainly produced by NADH oxidation via ETP under aerobic conditions [22]. As a result, we hypothesized that ATP availability is another limiting factor for further increasing α-farnesene production.

### 2.4. Overexpression of Adenine Phosphoribosyltransferase Enhances the Precursor AMP Supply and Thus Increases the ATP Availability and α-Farnesene Production in P. pastoris

ATP can be synthesized either by substrate level phosphorylation (SLP) or by ETP in aerobic respiring bacteria, and the ETP is the main route for ATP generation using NADH as an electron donor [22,43]. In the process of ETP, AMP or/and ADP are used as the substrate for ATP biosynthesis [22]. In theory, increasing the AMP or ADP supply could increase the ATP availability. To test this theory, the endogenous adenine phosphoribosyltransferase (APRT) was overexpressed in strain X33-35, resulting in strain X33-37. As a control, the endogenous APRT was also overexpressed in strain X33-30*, resulting in strain X33-30*A. APRT catalyzes the formation of AMP from adenine and 5-phospho-α-Dribose-1-diphosphate (PRPP) [44], and we found that the intracellular ATP level of strain X33-37 was 9.4% higher than that of strain X33-35, while the intracellular NADH level was slightly lower than that of strain X33-35. Strain X33-30*A showed the same trend as strain X33-37 in that the intracellular ATP level was higher and the intracellular NADH level was slightly lower than strain X33-30* (Table 1). Similar results were also found in previous reports, in which the mutated *Corynebacterium glutamicum* with the inactivation of AMP nucleosidase showed an increased intracellular ATP level and a decreased intracellular NADH level [20]. The most likely reason for this is that more NADH was used for ATP biosynthesis through ETC due to the abundant supply of AMP. Unsurprisingly, the overexpression of the *aprt* gene promoted cell growth and α-farnesene production. The DCW and α-farnesene production of strain X33-37 reached 2.35 ± 0.11 g/L and 2.94 ± 0.25 g/L (Figure 6), which were 10.3% and 6.1% higher than those of strain X33-35 (i.e., 2.13 ± 0.16 g/L and 2.77 ± 0.18 g/L, respectively), respectively. Strain X33-30*A also showed increased α-farnesene production as compared with strain X33-30* (i.e., 2.38 ± 0.18 g/L vs. 2.17 ± 0.15 g/L) (data not shown). ATP is a key factor for cell growth and maintenance and controlling the intracellular environment [45], and thus an adequate ATP supply could increase biomass. In addition, the biosynthesis of one molecule of α-farnesene requires at least nine molecules of ATP (Figure 1), so the increased intracellular ATP level effectively pulled more carbon flux into the α-farnesene biosynthetic pathway, resulting in higher α-farnesene production. These findings suggest that the overexpression of endogenous APRT promotes increased α-farnesene production by facilitating AMP biosynthesis and thus increasing ATP supply.

### 2.5. Deletion of NADH-Dependent Dihydroxyacetone Phosphate Reductase Elevates the Intracellular NADH Level and Thus Elevates the Intracellular ATP Level and α-Farnesene Production in P. pastoris

NADH plays an important role in maintaining the redox balance and energy generation of NADH [18]. It can used as precursor for the regeneration of NADPH and ATP. In order to maintain the abundant supply of NADH, we tried to decrease the NADH consumption in the shunt pathway. To do this, we discarded the NADH-dependent dihydroxyacetone phosphate reductase (GPD1) in strain X33-37, resulting in strain X33-38. As a control, the GPD1 in strain X33-30* was also inactivated, resulting in strain X33-30*∆ga. GPD1 catalyzes the biosynthesis of glycerol from dihydroxyacetone phosphate and uses NADH as a reducing cofactor (Figure 1). Previous research indicated that the glycerol biosynthetic pathway plays an important role in maintaining the intracellular NADH and NAD^+^ levels [46]. Therefore, the intracellular NADH level and NADH/NAD^+^ ratio in the strain X33-38 increased by 11.6% and 28.6% as compared with strain X33-37, respectively (Table 1). Meanwhile, strain X33-38 had improvements in both the intracellular NADPH level and the ATP level (Table 1). The activation of GPD1 was also increased the intracellular NADH/NAD^+^ ratio, the NADPH/NADP^+^ ratio and ATP in strain X33-30*∆ga (Table 1), indicating that the deletion of NADH-dependent GPD1 elevates the intracellular NADH level and ATP level. As a result, the α-farnesene production of strain X33-38 reached 3.09 ± 0.37 g/L after 72 h in shake flask fermentation, which was 5.1% higher than that of strain X33-37 (i.e., 2.94 ± 0.25 g/L) (Figure 7a). The DCW of strain X33-38 was also slightly increased as compared with strain X33-37 (i.e., 2.41 ± 0.23 g/L vs. 2.35 ± 0.11 g/L) (Figure 7b). Although strain X33-30*∆ga did not show a significant difference in cell growth as compared with strain X33-30* (Figure 7d), it showed a higher α-farnesene production than that of strain X33-30* (Figure 7e). It is worth noting that the GPD1-deficient strains (i.e., strains X33-38 and X33-30*∆ga) did not accumulate glycerol throughout the fermentation process, which was different from the strains X33-30* and X33-37 (Figure 7c,f). GPD1 is a key enzyme in glycerol biosynthesis [47] and He et al. [46] discovered similar results, in which the strain *S. cerevisiae* DRY01 with GPD1 silencing significantly reduced glycerol accumulation. These results show that removing GPD1 not only increases the amount of NADH, but also decreases the flow of carbon in the shunt pathway, which produces more α-farnesene.

## 3. Materials and Methods

### 3.1. Strains and Medium 

All engineered strains used in this study are based on the *P. pastoris* strain X33-30 [3]. *E. coli* JM109 strain was used to maintain and amplify plasmids, and recombinant strains were cultured at 37 °C in LB medium containing 50 mg/L kanamycin, 25 mg/L zeocin or 100 mg/L ampicillin. *P. pastoris*-engineered recombinant strains were cultivated in YPD medium at 30° and screened with 100 mg/L zeocin or 500 mg/L geneticin, recovery of selectable markers by the Cre/LoxP system. The rich YPD medium contained 20 g/L glucose, 20 g/L peptone and 10 g/L yeast extract. The medium YPDA for inducing Cre enzyme expression contained 20 g/L L-galactose, 10 g/L yeast extract and 20 g/L peptone. 

### 3.2. Construction of Plasmids and Strains

Some of the constructed strains and recombinant plasmids in this study were listed in Table 2, respectively. The designed primers were listed in Appendix A. The integration site of the strain was the P_GAP_ promoter site, or the *his4* site. The specific strain construction procedures were included in the “Appendix A”. The LoxP sites in the Cre/LoxP system were mutated into Lox71 and Lox66 sites, respectively, to prevent repeated cleavage and recombination by the Cre enzyme. The details of DNA manipulation and transformation were described in “Appendix A”.

### 3.3. Shake Flask Culture Conditions and Biomass Analysis

For activation, preserved strains were cultured in YPD medium at 30 °C. Then, the appropriate amount of activated strain was added into 10 mL of liquid medium overnight to become seed medium. The initial cells OD_600_ with 50 mL of YPD fermentation medium was 0.15. The pH of 100 mM/L potassium phosphate buffer was adjusted to 6.0, and then add into the fermentation broth, accounting for 10% of it. The upper layer was overlaid with 10% n-dodecane, and the fermentation was completed after 72 h in a reciprocating shaker at 30 °C with 100 rpm. OD_600_ of *P. pastoris* cells were measured using ultraviolet spectrophotometer (UV2100, Shanghai, China). Dry cell weight (DCW) was measured as described by Tomas et al. [11].

### 3.4. Quantification of α-Farnesene 

The fermentation broth was centrifuged at 12,000 rpm for 10 min. Then, the upper layer of n-dodecane liquid was filtered. A total of 100 μL of samples in n-dodecane phase were taken out and diluted ten times, and then a small amount of anhydrous copper sulfate was added to remove residual water. Quantification was performed using Shimadzu GC-2010 plus, a gas chromatograph equipped with a flame ionization detector (FID). The detection method of α-farnesene is as described by Liu et al. [3]. The standard α-farnesene, antibiotics and chemicals were purchased from Sigma (Sigma-Aldrich, Burlington, MA, USA).

### 3.5. Quantification of Intracellular NADH/NAD^+^, NADPH/NADP^+^ and ATP 

Strains were cultivated in YPD medium for 24 h or 72 h, and cells were harvested by centrifugation at 4 °C for 30 min at 10,000× *g* and re-suspended in cold PBS buffer to OD_600_ = 10. The intracellular NADH/NAD and NADPH/NADP were extracted according to the previously described method by Faijes et al. [48], and the intracellular ATP was extracted according to Ni et al.’s reports [49]. Their concentrations were measured using the NADH/NAD^+^ Quantification Colorimeteric Kit, NADPH/NADP^+^ Quantification Colorimeteric Kit and ATP Colorimetric/Fluorometric Assay Kit (BioVision, Inc., Milpitas, CA, USA) according to the manufacturers’ instructions.

### 3.6. Statistical Analysis

The experiments in this study were independently performed at least three times, and data are expressed as mean and standard deviation (±SD). Student’s *t* test was used to discuss the statistical difference among the experiment data.

## 4. Conclusions

α-Farnesene is biosynthesized from farnesyl pyrophosphate (FPP) and dimethylallyl pyrophosphate (DMAPP), which are mainly produced through the mevalonate (MVA) pathway in *P. pastoris* [3,6], and thus six molecules of NADPH and nine molecules of ATP are needed to produce one molecule of α-farnesene (Figure 1). Although many effective strategies have been used to increase α-farnesene production in yeast, for example, in *P. pastoris* [3], *Saccharomyces cerevisiae* [50,51] and *Yarrowia lipolytica* [2,52], these studies mainly focused on modifying the carbon’s metabolism pathway to enhance the carbon flux in the α-farnesene biosynthetic pathway. We first attempted to rationally reconstruct the NADPH and ATP biosynthetic pathways in order to increase α-farnesene production in an α-farnesene-producing strain, *P. pastoris* X33-30*. The resultant strain, *P. pastoris* X33-38, produced 3.09 ± 0.37 g/L of α-farnesene after 72 h in shake flask fermentation, which is the highest value ever reported to the best of our knowledge (Table 3).

In *P. pastoris*, oxiPPP is the main pathway for NADPH generation, but the overexpression of all genes in oxiPPP is not an adequate choice for increasing α-farnesene production due to the disturbance of the carbon flux in the PPP [24] and acetyl-CoA biosynthetic pathways [29]. The combined over-expression of ZWF1 and SOL3 improves the NADPH supply, thus increasing α-farnesene production. Furthermore, the intracellular NADPH level can be increased further through the heterologous expression of cPOS5 rather than PGI inactivation, resulting in an α-farnesene production of 2.77 ± 0.18 g/L in strain X33-35. Increasing the ATP supply, as the other key cofactor in α-farnesene production, also plays an important role in promoting α-farnesene production. The strain X33-38, with the overexpression of APRT and the deletion of GPD1 to increase the supply of AMP and NADH for ATP generation, exhibits visibly increased cell growth and α-farnesene production. Therefore, the rational modification of the NADPH and ATP regeneration pathways plays a vital role in facilitating α-farnesene biosynthesis in *P. pastoris*. These results also provide new directions and references for constructing strains for NADPH and/or ATP-dependent production of high-added value products.

## Figures and Tables

**Figure 1 ijms-24-01767-f001:**
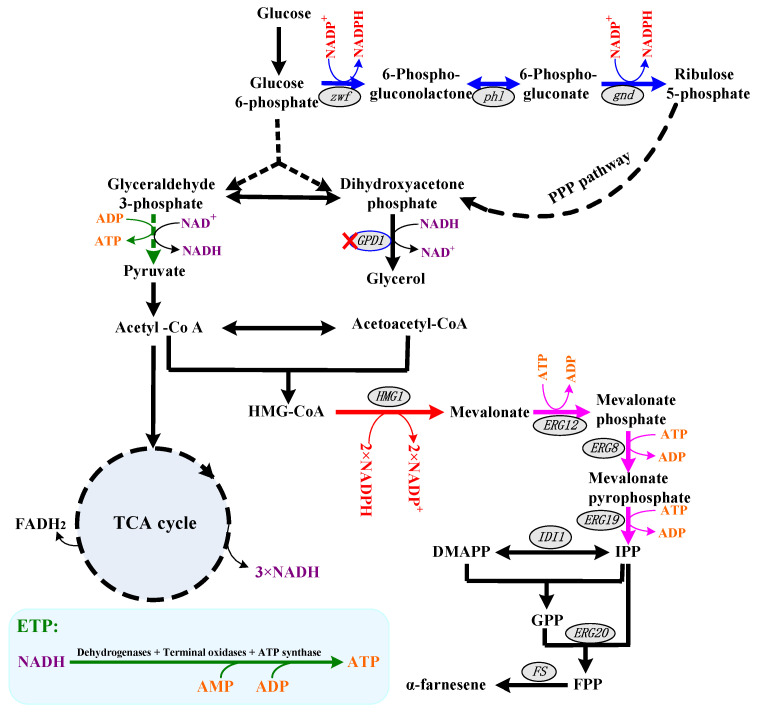
The schematic diagram of α-farnesene biosynthetic pathways with NADPH and ATP regeneration pathway in *Pichia pastoris*. The pathways of NADPH biosynthesis are shown in blue lines, and the pathways of NADPH catabolism are shown in red lines. The pathways of ATP biosynthesis are shown in green lines, and the pathways of ATP catabolism are shown in pink lines. The key genes are listed in ellipses. Abbreviations: HMG-CoA, hydroxymethylglutaroyl coenzyme A; IPP, isopentenyl pyrophosphate; DMAPP, dimethylallyl pyrophosphate; GPP, geranyl pyrophosphate; FPP, farnesyl pyrophosphate.

**Figure 2 ijms-24-01767-f002:**
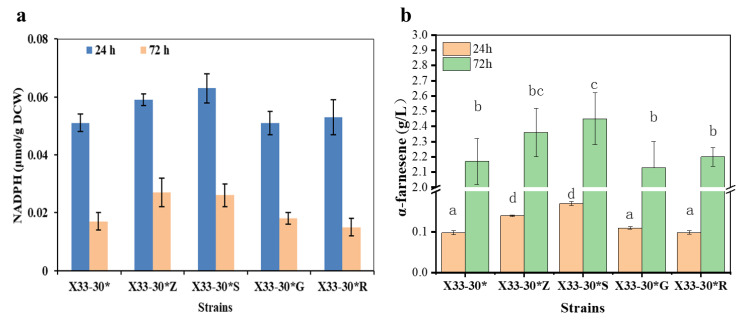
Screening the best enzyme in oxiPPP for α-farnesene biosynthesis. (**a**) Overexpression of the key enzymes in oxiPPP affects the intracellular NADPH level. (**b**) The effects of overexpression of the key enzymes in oxiPPP on α-farnesene production after 72 h cultivation. The strain X33-30* was used as the control (grey bar). These data represent average values and standard deviations achieved from at least three independent experiments. Different letters indicate statistically significant differences (*p* < 0.05).

**Figure 3 ijms-24-01767-f003:**
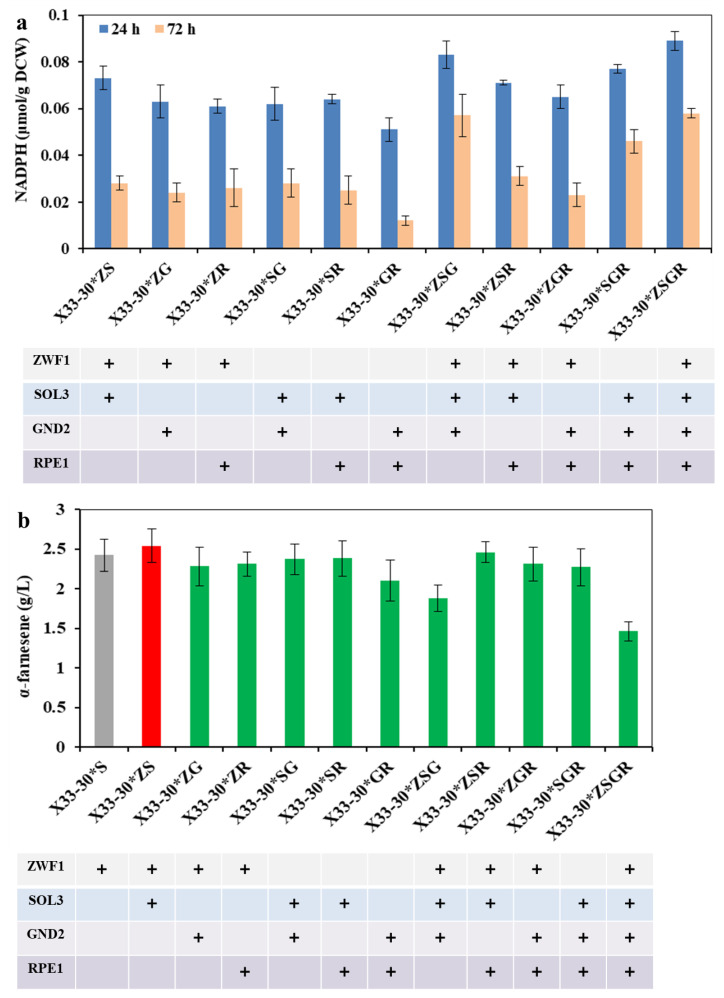
Optimization of the expression combination ways of the enzymes in oxiPPP for α-farnesene biosynthesis. (**a**) The effects of the different expression combination ways on the intracellular NADPH level. (**b**) The effects of the different expression combination ways on α-farnesene production after 72 h of cultivation. The strain X33-30*S was used as the control (grey bar), and the best strain is shown by the red bar.

**Figure 4 ijms-24-01767-f004:**
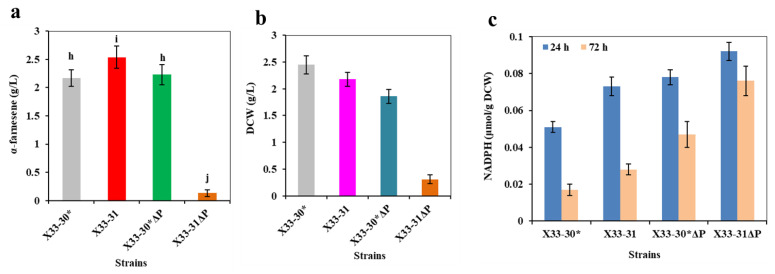
Inactivation of PGI1 negatively affects α-farnesene biosynthesis. (**a**) α-Farnesene titers. (**b**) Dry cell weight (DCW). (**c**) The intracellular NADPH level at 24 h and 72 h. The strains X33-30* and X33-31 were used as the control (grey bar). These data represent average values and standard deviations achieved from at least three independent experiments. Different letters indicate statistically significant differences (*p* < 0.05).

**Figure 5 ijms-24-01767-f005:**
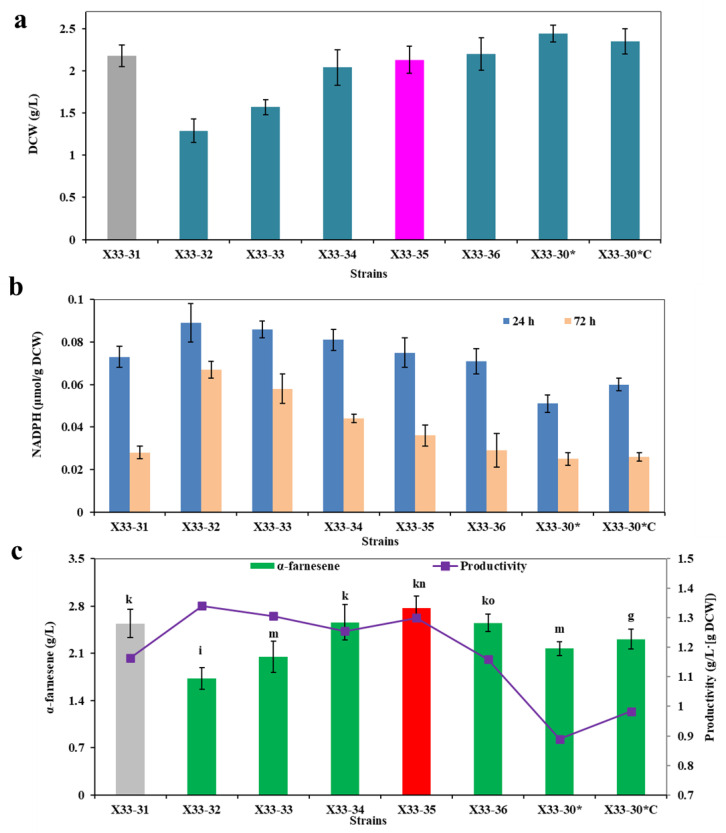
Optimization of the expression level of cPOS5 for α-farnesene biosynthesis. (**a**) Dry cell weight (DCW). (**b**) The intracellular NADPH level at 24 h and 72 h. (**c**) α-Farnesene titers and productivity. The strain X33-31 was used as the control (grey bar). These data represent average values and standard deviations achieved from at least three independent experiments. Different letters indicate statistically significant differences (*p* < 0.05).

**Figure 6 ijms-24-01767-f006:**
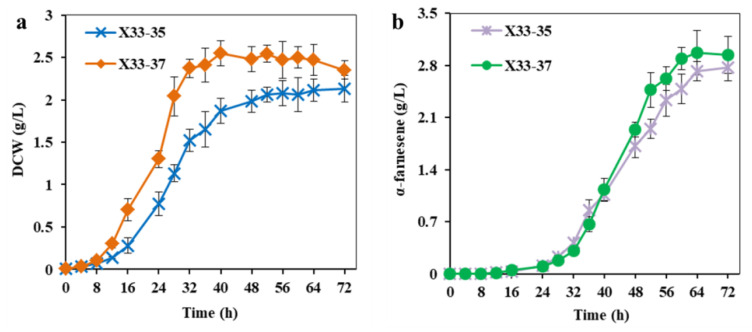
Overexpression of APRT to increase ATP supply for α-farnesene production. (**a**) The cell growth of strain X33-37 with overexpression of APRT. (**b**) The α-farnesene production of strain X33-37 with overexpression of APRT. The strain X33-35 without overexpression of APRT was used as the control. These data represent average values and standard deviations achieved from at least three independent experiments.

**Figure 7 ijms-24-01767-f007:**
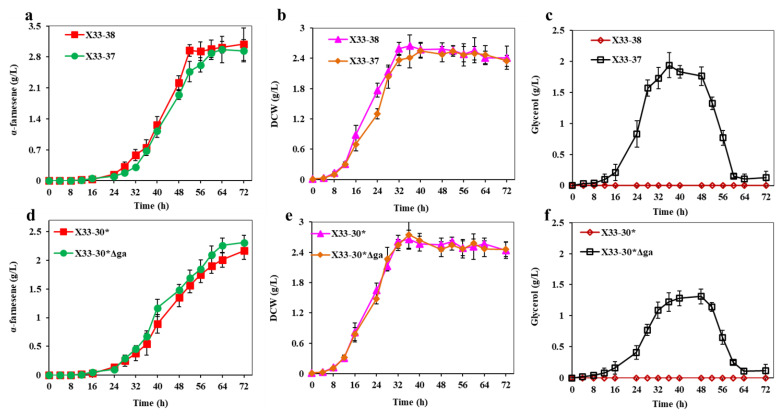
Inactivation of GPD1 to decrease the NADH consumption in shunt pathway. (**a**,**e**) α-farnesene titers of strains. (**b**,**d**) DCW of strains. (**c**,**f**) Glycerol titers of strains. These data represent average values and standard deviations achieved from at least three independent experiments.

**Table 1 ijms-24-01767-t001:** Comparison of intracellular nucleotides concentrations in *P. pastoris* strains (μmol/(g DCW)) ^a^.

Strains	NADH	NAD^+^	NADH/NAD^+^	NADPH	NADP^+^	NADPH/NADP^+^	ATP
X33-30*	4.92 ± 0.10	11.32 ± 0.53	0.43	0.051 ± 0.003	0.213 ± 0.015	0.24	5.93 ± 0.26
X33-30*A	4.81 ± 0.23	11.83 ± 1.05	0.41	0.058 ± 0.004	0.215 ± 0.010	0.27	6.28 ± 0.22
X33-30*∆ga	5.16 ± 0.20	11.05 ± 1.05	0.47	0.064 ± 0.004	0.201 ± 0.010	0.32	6.20 ± 0.30
X33-31	4.68 ± 0.35	12.03 ± 1.12	0.39	0.073 ± 0.005	0.194 ± 0.017	0.38	5.19 ± 0.45
X33-35	3.19 ± 0.24	13.76 ± 1.24	0.23	0.075 ± 0.007	0.186 ± 0.009	0.40	3.07 ± 0.42
X33-37	2.93 ± 0.31	14.21 ± 1.38	0.21	0.073 ± 0.009	0.199 ± 0.015	0.37	3.36 ± 0.23
X33-38	3.27 ± 0.04	13.63 ± 1.09	0.27	0.077 ± 0.004	0.186 ± 0.018	0.41	4.03 ± 0.37

^a^ The cells cultured in YPD medium for 24 h in shake flasks were used for analysis. All data are meaning values of at least three determinations of at least three independent experiments with ± SD.

**Table 2 ijms-24-01767-t002:** The main strains used in the study.

*P. pastoris*	Characteristics	Reference
X33-30	An α-farnesene-producing strain derived from *P. pastoris* X33 by dual regulation of the carbon’s metabolic pathways in cytoplasm and peroxisomes	[3]
X33-30*	In the X33-30 strain, P_CAT1_ promoters were replaced with P_GAP_ promoters.	This study
X33-31	Strain X33-30* with overexpression of ZWF1 and SOL3	This study
X33-32	Strain X33-31 with overexpression of cPOS5 under control by promoter P_GAP_	This study
X33-33	Strain X33-31 with overexpression of cPOS5 under control by promoter P_PISI_	This study
X33-34	Strain X33-31 with overexpression of cPOS5 under control by promoter P_GPM1_	This study
X33-35	Strain X33-31 with overexpression of cPOS5 under control by promoter P_MET3_	This study
X33-36	Strain X33-31 with overexpression of cPOS5 under control by promoter P_KEX2_	This study
X33-37	Strain X33-35 with overexpression of APRT	This study
X33-38	Strain X33-37 with inactivation of GPD1	This study
pGAPZA	Zeocin^r^, expression vector with GAP promoter	Invitrogen
pPIC3.5k	Kana ^r^, expression vector	Invitrogen
pGAP-Z	Zeocin^r^, pGAPZA plasmid with mutated LoxP site and his4 gene	This study
pPISI-Z	Replacing the GAP promoter with the PISI promoter in the pGAP-Z plasmid	This study
pGPM1-Z	Replacing the GAP promoter with the GPM1 promoter in the pGAP-Z plasmid	This study
pMET3-Z	Replacing the GAP promoter with the MET3 promoter in the pGAP-Z plasmid	This study
pPGK1-Z	Replacing the GAP promoter with the PGK1 promoter in the pGAP-Z plasmid	This study
pGAP-1	pGAP-Z carrying gene *zwf1*	This study
pGAP-2	pGAP-Z carrying gene *sol3*	This study
pGAP-3	pGAP-Z carrying gene *gnd2*	This study
pGAP-4	pGAP-Z carrying gene *rpe1*	This study
pGAP-5	pGAP-Z carrying gene *cPOS5*	This study
pPISI-1	pPISI-Z carrying gene *cPOS5*	This study
pGPM1-1	pGPM1-Z carrying gene *cPOS5*	This study
pMET3-1	pMET3-Z carrying gene *cPOS5*	This study
pPGK1-1	pPGK1-Z carrying gene *cPOS5*	This study
pAPRT-1	pGAP-Z carrying gene *aprt*	This study
pCas9-PG1-sg	pPIC3.5k plasmid with P_THX1_ promoter, cas9 and PG1-sgRNA. Additionally, the *his*4 gene was replaced with the ARS gene of *P. pastoris*	This study
pCas9-GPD1-sg	pPIC3.5k plasmid with P_THX1_ promoter, cas9 and GPD1-sgRNA. Additionally, the *his*4 gene was replaced with the ARS gene of *P. pastoris*	This study

**Table 3 ijms-24-01767-t003:** Overview on the production of α-farnesene.

Strains	Culturing Methods	Carbon Source	Final Titers (g/L)	Productivity (g/L/h) ^a^	References
*P. pastoris* X33-38	Shake flasks	Glucose	3.09	0.043	This work
*S. cerevisiae* WH62S	Shake flasks	Glucose	1.48	0.009	[50]
	Fed-batch	Glucose	10.4	0.043	
*P. pastoris* X33-30	Shake flasks	Sorbitol + Oleic acid	2.56	0.036	[3]
*Synechococcus elongatus* SeHL-FN03	Shake flasks	CO_2_	5.0 × 10^−3^	2.604 × 10^−5^	[53]
*Yarrowia lipolytica LSC28*	Shake flasks	Glycerol	9.0 × 10^−2^	7.500 × 10^−4^	[2]
	Fed-batch		2.57	0.021	
*C. glutamicum* JP-2	48-well plates	Glucose	0.28	NA ^b^	[54]
*Y. lipolytica* F5	Shake flasks	Glucose	1.70	5.903×10^−3^	[6]
	Fed-batch		25.55	0.089	

^a^ Estimated from reference. ^b^ NA: unavailable.

## Data Availability

The data presented in this study are available upon request from the corresponding author.

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
