# Peer review of "Cofactor Engineering for Efficient Production of α-Farnesene by Rational Modification of NADPH and ATP Regeneration Pathway in Pichia pastoris"

_ijms, 2023, doi:10.3390/ijms24021767_

Round 1

Reviewer 1 Report

Information in this paper is of great interest considering the higher need for new chemical platforms compounds. Nature has a great capacity of producing biomolecules, but is our duty to find ways to produce them at a larger scale. 

The research part is well done, but the way is it presented in the manuscript demonstrates that the authors did not took enough time to write it. I appreciate the wok done and the science, but there are a lot of phases that are not complete, some of then did not have any meaning. So I will kindly ask the authors to take time and re-write parts of the manuscript. I marked in yellow some of them but are still many more. 

Author Response

1) Information in this paper is of great interest considering the higher need for new chemical platforms compounds. Nature has a great capacity of producing biomolecules, but is our duty to find ways to produce them at a larger scale.

  • Response 1: Thank you very much for your recognition of this work, and we agree with the view that "nature has a great capacity of producing biomolecules, but is our duty to find ways to produce them at a large scale".

2) The research part is well done, but the way is it presented in the manuscript demonstrates that the authors did not took enough time to write it. I appreciate the wok done and the science, but there are a lot of phases that are not complete, some of then did not have any meaning. So I will kindly ask the authors to take time and re-write parts of the manuscript. I marked in yellow some of them but are still many more.

  • Response 2: Thank you very much for your recognition of this work, and sorry for the trouble we brought you due to our carelessness. According to the suggestions of reviewers, we have seriously revised the manuscript to organize our ideas in this manuscript and to show results for the readers. In particular, the described method has been revised and the revised part has been marked in red in the manuscript. In addition, this manuscript has been edited by professional editors at Editage (Shanghai, China), who provides English-improvement services. Moreover, we have now read through this manuscript again and we are sure that there are no longer any spelling mistakes (In red font). Therefore, I think the quality of the English is satisfactory for this journal.

Reviewer 2 Report

This study improved the production of α-farnesene by the resultant strain of Pichia pastoris X33-30 which was obtained in the previous study conducted by the same research group. The research team focused on very interested research point related to increase the supply of NADPH and ATP as developing keys for a α-farnesene high-producing strain.

They found that the overexpression of essential enzymes of  ZWF1 and SOL3 improved NADPH supply and subsequently increased the production of α-farnesene without inactivation of glucose 6-phosphate isomerase (PGI) pathway, with any negative effect on cell growth. They introduced the oxidative branch of pentose phosphate pathway (oxiPPP) into Pichia pastoris X33-30 to improved NADPH supply. Improvement of α-farnesene high-producing strain was also conducted by increasing ATP availability which was attempted by increasing the supply of adenosine monophosphate (AMP) for ATP synthesis and decreasing NADH consumption in the shunt pathway. The subject is very interested; however, my major comments are listed below:

In Fig 2b, From the presented data, there were no any significant differences in α-farnesene concentrations within different expressed strains compared to strain X33-30*. This means that there was no significant effect of the selected overexpressed the single genes on α-farnesene concentrations. Also, authors should conduct a statistical analysis of these data.

In Fig.2. Authors need to explain why NADPH levels were significantly reduce after 72h compared to after 24. Also, to better link between NADPH levels and α-farnesene production (fig2a and fig2b), the concentration of α-farnesene need to be estimated after 24h.

In general, Fig 2 should be improved, and statistically letters need to be added.

After the overexpressed the single genes, authors tried the synergetic effects of some key genes in oxiPPP on NADPH and α-farnesene production. Authors reported that in L168-169 and based on the results presented in Fig. 3. “As a result, the strain X33-31 was chosen to be further modified in order to increase α-farnesene production.” However, again there was no significant effect of combined expressed genes in oxiPPP on α-farnesene production. Also, if the authors compare between the results presented in Figs 2 and 3, they will find that there are no differences in the α-farnesene production between the expressed strains in the overexpressed the single genes nor in the combined genes. So, choice the stain X33-31 for further modification to increase the α-farnesene production based on negative response is significantly affected the accuracy of the next results and conclusion.

In section, 2.2. Inactivation of glucose-6-phosphate isomerase disturbs the cell growth and thus negatively  affecting α-farnesene biosynthesis in P. pastoris X33-31. Authors concluded that inactivation of PGI1 negatively affects α-farnesene biosynthesis, they should be carful about this very important conclusion. The presented results in this part showed that inactivation of PGI did not significantly affect α-farnesene biosynthesis on strains X33-30* and X33-30*ΔP. Also, the linked results between α-farnesene concentration and dry cell weight (DCW, cell growth) should be interpreted. For example, strain X33-30*ΔP produced lower DCW than X33-30*, however this strain produced higher farnesene.

In section 2.3. Low intensity expression of POS5 from S. cerevisiae balances the NADPH/NADH ratio and thus promoting α-farnesene biosynthesis in P. pastoris. In order to link between low intensity of POS5 and levels of NADPH/NADH and α-farnesene biosynthesis, the experiment should be conducted to introduce the cPOS5 targeting in the cytosol from S. cerevisiae in both strains of X33-30* and X33-31, not only strain 31.

By similar way, all modification in section 2.4. and 2.5. the experiments of Overexpression of adenine phosphoribosyl transferase and Deletion of NADH-dependent dihydroxyacetone phosphate reductase  should be conducted on strain X33-30*.

Unfortunately, this study showed a very weak results in term of low α-farnesene production and low yield by the utilized strains under study.

Statistical analysis should be done and presented for all obtained results.

Author Response

1) This study improved the production of α-farnesene by the resultant strain of Pichia pastoris X33-30 which was obtained in the previous study conducted by the same research group. The research team focused on very interested research point related to increase the supply of NADPH and ATP as developing keys for a α-farnesene high-producing strain.They found that the overexpression of essential enzymes of ZWF1 and SOL3 improved NADPH supply and subsequently increased the production of α-farnesene without inactivation of glucose 6-phosphate isomerase (PGI) pathway, with any negative effect on cell growth. They introduced the oxidative branch of pentose phosphate pathway (oxiPPP) into Pichia pastoris X33-30 to improved NADPH supply. Improvement of α-farnesene high-producing strain was also conducted by increasing ATP availability which was attempted by increasing the supply of adenosine monophosphate (AMP) for ATP synthesis and decreasing NADH consumption in the shunt pathway. The subject is very interested; however, my major comments are listed below.

  • : Thank you very much for your recognition of this work and your major comments.

2) In Fig 2b, From the presented data, there were no any significant differences in α-farnesene concentrations within different expressed strains compared to strain X33-30*. This means that there was no significant effect of the selected overexpressed the single genes on α-farnesene concentrations. Also, authors should conduct a statistical analysis of these data.

  • : Thank you for your comment. And you are right, it doesn't seem to be any significant differences in α-farnesene concentrations within different expressed strains compared to strain X33-30*. However, as can be seen from Fig 2b, strains X33-30*Z and X33-30*S increased α-farnesene production by about 6.5% and 12.0% as compared with strain X33-30* at 72 h, respectively. Based on previous reports, the production of the strain has been steadily increased by 12% in the shake flask fermentation, indicating that the overexpression gene has a positive effect on it (Zhang X, et al. Combinatorial engineering of Saccharomyces cerevisiae for improving limonene production. Biochemical Engineering Journal, 2021, doi:10.1016/J.BEJ.2021.108155; Liu YH, et al. Engineering the oleaginous yeast Yarrowia lipolytica for production of α-farnesene. Biotechnol Biofuels, 2019, 12:296.). In addition the α-farnesene yield of strain X33-30*S also showed the significantly different from the strain X33-30* based on systematic analysis. According to the suggestions of reviewers, we conducted a statistical analysis of these data (Fig 2b). I hope my opinion and interpretation will not affect your decision.

3) In Fig.2. Authors need to explain why NADPH levels were significantly reduce after 72h compared to after 24. Also, to better link between NADPH levels and α-farnesene production (fig2a and fig2b), the concentration of α-farnesene need to be estimated after 24h.

  • : Thank you for your comment. The reviewers are right. The NADPH levels were significantly reduced after 72 h as compared to after 24 h. In our opinions, because 6 molecules of NADPH and 9 molecules of ATP are needed to produce 1 molecule of α-farnesene, α-farnesene synthesis is essentially in the middle to late stage and thus NADPH is consumed in large amounts. According to the suggestions of reviewers, we also added the relevant explanations in the article marked in red font. I hope my opinion and interpretation will not affect your decision.

4) In general, Fig 2 should be improved, and statistically letters need to be added.

  • : Thank you for your comments and suggestions. According to the suggestions of reviewers, we have added the statistical analysis results in Fig 2.

5) After the overexpressed the single genes, authors tried the synergetic effects of some key genes in oxiPPP on NADPH and α-farnesene production. Authors reported that in L168-169 and based on the results presented in Fig. 3. “As a result, the strain X33-31 was chosen to be further modified in order to increase α-farnesene production.” However, again there was no significant effect of combined expressed genes in oxiPPP on α-farnesene production. Also, if the authors compare between the results presented in Figs 2 and 3, they will find that there are no differences in the α-farnesene production between the expressed strains in the overexpressed the single genes nor in the combined genes. So, choice the stain X33-31 for further modification to increase the α-farnesene production based on negative response is significantly affected the accuracy of the next results and conclusion.

  • : Thank you for your comments. In Figure 2, we can see that the α-farnesene yield of strain X33-30*S with overexpress sol3 showed the significantly different from that of strain X33-30* based on systematic analysis. In addition, we also can clearly see that the α-farnesene yield of strain X33-31 is the highest, and the combination of overexpression of corresponding genes also obviously changed the α-farnesene yield of strains (Fig. 3). And you are right, some combination overexpression really has no effect. For this, we also pointed out in the article and made corresponding analysis on lines 149 to 165. In Figure 4, we can see that the yield of α-farnesene in X33-31 and X33-30* is significantly different, indicating that the combination of overexpression of ZWF1 and SOL3 promotes the synthesis of α-farnesene. These results indicated that moderate increase of NADPH concentration would be beneficial to α-farnesene synthesis. So, choice the stain X33-31 for further modification to increase the α-farnesene production is rational, both in the α-farnesene synthesis ability of the strain and in the sufficient amount of NADPH in the cell. None of this affects some of the next conclusions and results. I hope my opinion and interpretation will not affect your decision.

6) In section, 2.2. Inactivation of glucose-6-phosphate isomerase disturbs the cell growth and thus negatively affecting α-farnesene biosynthesis in P. pastoris X33-31. Authors concluded that inactivation of PGI1 negatively affects α-farnesene biosynthesis, they should be careful about this very important conclusion. The presented results in this part showed that inactivation of PGI did not significantly affect α-farnesene biosynthesis on strains X33-30* and X33-30*ΔP. Also, the linked results between α-farnesene concentration and dry cell weight (DCW, cell growth) should be interpreted. For example, strain X33-30*ΔP produced lower DCW than X33-30*, however this strain produced higher farnesene.

  • : Thanks for your reminder. As can be seen from Fig. 4, inactivation of PGI in P. pastoris X33-31 reduced the α-farnesene production and cell growth (DCW). The reviewers are right, inactivation of PGI did not significantly affect α-farnesene biosynthesis on strains X33-30* and X33-30*ΔP but it significantly inhibited the cell growth (Fig. 4b). Many previous reports indicated that inactivation of PGI in yeast was not be helpful to cell growth (Qin N, et al. Rewiring central carbon metabolism ensures increased provision of acetyl-CoA and NADPH required for 3-OH-propionic acid production. ACS Synth Biol, 2020, 9 (12): 3236-3244; Zhang Q, et al. Metabolic engineering of Pichia pastoris for myo-inositol production by dynamic regulation of central metabolism. Microb Cell Fact 2022, 21(1): 112.). However, there's no denying that inactivation of PGI enforced the carbon flux into PPP, thus increasing the NADPH availability for α-farnesene biosynthesis. And this is why the strain X33-30*ΔP with inactivation of PGI showed the no significant decrease in α-farnesene biosynthesis despite of the significant decrease in cell growth. However, there may be surplus NADPH in strain X33-31ΔP because it performed over-expression of ZWF1 and SOL3 and inactivation of PGI to improve the NADPH supply. Heux et al. (Heux S, et al. Glucose utilization of strains lacking PGI1 and expressing a transhydrogenase suggests differences in the pentose phosphate capacity among Saccharomyces cerevisiae strains. FEMS Yeast Res, 2008, 8(2): 217-24.) pointed out that the PGI-deficient strain did not grow on glucose because the surplus NADPH cannot be re-oxidized. According to the suggestions of reviewers and to more rigorously explain the thesis point, the conclusion that inactivation of PGI1 negatively affects α-farnesene biosynthesis was limited in strain X33-31. The details are listed in the main manuscript and have been marked in red in the manuscript. I hope my opinion and interpretation will not affect your decision.

7) In section 2.3. Low intensity expression of POS5 from S. cerevisiae balances the NADPH/NADH ratio and thus promoting α-farnesene biosynthesis in P. pastoris. In order to link between low intensity of POS5 and levels of NADPH/NADH and α-farnesene biosynthesis, the experiment should be conducted to introduce the cPOS5 targeting in the cytosol from S. cerevisiae in both strains of X33-30* and X33-31, not only strain 31.

  • : Thanks for the reviewer's suggestion. And the reviewers are right, introduction of the cPOS5 targeting in the cytosol from S. cerevisiae in both strains of X33-30* and X33-31 could preferably illustrate the link between low intensity of POS5 and levels of NADPH/NADH and α-farnesene biosynthesis. However, due to our negligence in the experiment and the time problem at present we could not do that, and we think that this decision did not affect the accuracy of the next results and conclusion. On the one hand, the link between low intensity of POS5 and levels of NADPH/NADH and α-farnesene biosynthesis has been showed in strain X33-3 (Fig. 5), which is of reference value to moderately increase the intracellular NADPH of the strain and increase the production of related products. On the other hand, similar results can also be referred to in the following literature: [1] Tomàs Gamisans Màrius, et al."Redox Engineering by Ectopic Overexpression of NADH Kinase in Recombinant Pichia pastoris (Komagataella phaffii): Impact on Cell Physiology and Recombinant Production of Secreted Proteins." Applied and Environmental Microbiology 86.6(2019). doi:10.1128/AEM.02038-19; [2] Yukawa Takahiro, et al. "Optimization of 1,2,4-butanetriol production from xylose in Saccharomyces cerevisiae by metabolic engineering of NADH/NADPH balance." Biotechnology and bioengineering 118.1(2020). doi:10.1002/bit.27560. I hope my opinion and interpretation will not affect your decision.

8) By similar way, all modification in section 2.4. and 2.5. the experiments of Overexpression of adenine phosphoribosyl transferase and Deletion of NADH-dependent dihydroxyacetone phosphate reductase should be conducted on strain X33-30*.

  • : Thank you for your suggestion. And the reviewers are right, overexpression of adenine phosphoribosyl transferase and deletion of NADH-dependent dihydroxyacetone phosphate reductase in strain X33-30* resulted in the more rigorous experiments and the impeccable conclusions. As shown from our experimental results, overexpression of adenine phosphoribosyl transferase and deletion of NADH-dependent dihydroxyacetone phosphate reductase in strain X33-35 and X33-37, respectively, has certain reference value, and the experimental results are somewhat stable. We think these results without overexpression of adenine phosphoribosyl transferase and deletion of NADH-dependent dihydroxyacetone phosphate reductase in strain X33-30* did not affect the accuracy of the next results and conclusion. I hope my opinion and interpretation will not affect your decision.

9) Unfortunately, this study showed a very weak results in term of low α-farnesene production and low yield by the utilized strains under study.

  • : Thank you for your comments. The resultant strain P. pastoris X33-38 produced 3.09±0.37 g/L of α-farnesene after 72 h in shake-flask fer-mentation, which was 41.7% higher than that of the parent strain X33-38 (Liu H., et al. Dual regulation of cytoplasm and peroxisomes for improved α-farnesene production in recombinant Pichia pastoris. ACS Synth Biol, 2021, 10(6): 1563-1573.). As far as we know, the yield of α-farnesene in shake flask fermentation can be increased to 3.09±0.37 g/L, and the transformation result has been extremely significant. In addition, this is the highest value ever reported as we know it (Table 4). Although the titer of α-farnesene is far to production in industry, strain P. pastoris X33-38 is a competitive platform strain for α-farnesene production.

10) Statistical analysis should be done and presented for all obtained results.

  • : According to the suggestions of reviewers, we supplemented the methods of statistical analysis in the section of “3.6. Statistical analysis” and performed the statistical analysis in Figures (In red font).

Round 2

Reviewer 2 Report

I would like to thank the authors for their efforts to do this study. However, my major comment is about the choice the stain X33-31 for further modification to increase the α-farnesene production based on negative response is significantly affected the accuracy of the next results and conclusion for the details I presented in the first round of my revision. Consequently, the experiments should be conducted in both strains of X33-30* and X33-31, not only strain 31.

In the revised version, all the statistical analyses were at significant level of (p < 0.1), which means there are no significant effects.

Author Response

1) I would like to thank the authors for their efforts to do this study. However, my major comment is about the choice the stain X33-31 for further modification to increase the α-farnesene production based on negative response is significantly affected the accuracy of the next results and conclusion for the details I presented in the first round of my revision. Consequently, the experiments should be conducted in both strains of X33-30* and X33-31, not only strain 31.

  • : Thanks for your suggestion and your major comments. Your major comment is about the choice the stain X33-31 for further modification to increase the α-farnesene production based on negative response is significantly affected the accuracy of the next results. We paid a lot of attention to it. In Figure. 2b, we re-measured several groups of data, and found that there was a significant difference between strain X33-30*S and strain X33-30* in α-farnesene concentration (p<0.05). The content of α-farnesene in strain X33-31 was higher than that in strain X33-30*S, which was obviously significantly different from that in strain X33-30* (Figure 3a). Right for the next change under strainX33-31 base, as this will help us to obtain a α-farnesene high-producing strain. For the preciseness of our experiment, we also modified the strain X33-30*. In order to link between low intensity of POS5 and levels of NADPH/NADH and α-farnesene biosynthesis, the experiment have been conducted to introduce the cPOS5 targeting in the cytosol from S. cerevisiae in both strains of X33-30* and X33-31 (Figure 5). The experiments of overexpression of adenine phosphoribosyl transferase and deletion of NADH-dependent dihydroxyacetone phosphate reductase also have been conducted on strain X33-30*. The corresponding results and analysis have been marked in red in the manuscript.

2) In the revised version, all the statistical analyses were at significant level of (p < 0.1), which means there are no significant effects.

  • : We are very sorry to use p < 0.1, which makes it impossible to show the significant difference between our data. At present, we have carefully carried out the data statistical analysis including increasing the numbers of the independent experiments,and relevant significant differences have been added in the manuscript (p < 0.05) (In red font).

Round 3

Reviewer 2 Report

The revised version was improved. However, my major comment is about the choice the stain X33-31 for further modification to increase the α-farnesene production based on negative response is significantly affected the accuracy of the next results and conclusion for the details I presented in the first round of my revision. Consequently, the experiments should be conducted in both strains of X33-30* and X33-31, not only strain 31.

Author Response

1) The revised version was improved. However, my major comment is about the choice the stain X33-31 for further modification to increase the α-farnesene production based on negative response is significantly affected the accuracy of the next results and conclusion for the details I presented in the first round of my revision. Consequently, the experiments should be conducted in both strains of X33-30* and X33-31, not only strain 31.

  • Rep: Thanks again for your comments. We attach great importance to the questions you have raised, and we think that your suggestions can improve our experimental results. Therefore, the experiments suggested by you had been conducted in both strains of X33-30* and X33-31 in the second round revised manuscript. In Figure 2 (Page 4, Line 144; in red font), we added three groups of independent experiments to the original strains and compared them with the data obtained previously, indicating that the NADPH value and α-farnesene yield obtained were basically consistent with the previous independent experimental results. Statistical analysis of these data also found that there was a significant difference in α-farnesene production between strain X33-30*S and strain X33-30* (p<0.05)(Figure 2). As for the reason for selecting strain X33-31 for further modification, we have carried out corresponding analysis and description in Lines 174 to 177 on Page 5. In addition, cPOS5 from Saccharomyces cerevisiae was also overexpressed in strain X33-30*, and the data are shown in Figure 5. The corresponding analysis was listed in Lines 247 to 252 on Page 8 (in red font). Moreover, adenine phophoribosyltransfer was also overexpressed in strain X33-30*, and the corresponding results are shown in Table 3. The corresponding analysis was listed in Lines 280 to 292 (in red font). Deletion of NADH-dependent dihydroxyacetone phosphate reductase was also conducted in strain X33-30*, and the corresponding results are shown in Figure 7. The corresponding analysis was listed in Lines 316 to 338 on Page 11 (in red font). I hope my opinion will not affect your decision.